# Mapping the immune environment in clear cell renal carcinoma by single-cell genomics

Nicholas Borcherding[1,2,3,11], Ajaykumar Vishwakarma [4,5,6,11], Andrew P. Voigt[2], Andrew Bellizzi[7], Jacob Kaplan [7], Kenneth Nepple[3,8], Aliasger K. Salem [3,4], Russell W. Jenkins [5,6,12✉], Yousef Zakharia [3,8,9,12✉] & Weizhou Zhang [3,7,10,12✉]

Clear cell renal cell carcinoma (ccRCC) is one of the most immunologically distinct tumor types due to high response rate to immunotherapies, despite low tumor mutational burden. To characterize the tumor immune microenvironment of ccRCC, we applied single-cell-RNA sequencing (SCRS) along with T-cell-receptor (TCR) sequencing to map the transcriptomic heterogeneity of 25,688 individual CD45$^+$ lymphoid and myeloid cells in matched tumor and blood from three patients with ccRCC. We also included 11,367 immune cells from four other individuals derived from the kidney and peripheral blood to facilitate the identification and assessment of ccRCC-specific differences. There is an overall increase in CD8$^+$ T-cell and macrophage populations in tumor-infiltrated immune cells compared to normal renal tissue. We further demonstrate the divergent cell transcriptional states for tumor-infiltrating CD8$^+$ T cells and identify a *MKI67*+ proliferative subpopulation being a potential culprit for the progression of ccRCC. Using the SCRS gene expression, we found preferential prediction of clinical outcomes and pathological diseases by subcluster assignment. With further characterization and functional validation, our findings may reveal certain subpopulations of immune cells amenable to therapeutic intervention.

[1] Department of Pathology and Immunology, Washington University School of Medicine, St Louis, MO, USA. [2] Medical Science Training Program, University of Iowa, Iowa City, IA, USA. [3] Holden Comprehensive Cancer Center, University of Iowa, Iowa City, IA, USA. [4] Department of Pharmaceutical Sciences and Experimental Therapeutics, College of Pharmacy, University of Iowa, Iowa City, IA, USA. [5] Laboratory of Systems Pharmacology, Harvard Program in Therapeutic Science, Harvard Medical School, Boston, MA, USA. [6] Department of Medicine, Massachusetts General Hospital Cancer Center, Harvard Medical School, Boston, MA, USA. [7] Department of Pathology, University of Iowa Hospitals and Clinics, Iowa City, IA, USA. [8] Department of Urology, University of Iowa Hospitals and Clinics, Iowa City, IA, USA. [9] Department of Internal Medicine, University of Iowa Hospitals and Clinics, Iowa City, IA, USA. [10] Department of Pathology, Immunology, and Laboratory Medicine, University of Florida, Gainesville, FL, USA. [11]These authors contributed equally: Nicholas Borcherding, Ajaykumar Vishwakarma. [12]These authors jointly supervised this work: Russell W. Jenkins, Yousef Zakharia, Weizhou Zhang. ✉email: rjenkins@mgh.harvard.edu; yousef-zakharia@uiowa.edu; zhangw@ufl.edu

ccRCC is the most common type of renal cell carcinoma, comprising more than 70% of all renal cancers[1]. ccRCC represents an immune sensitive tumor type and is known for early advances in systemic immunotherapy using T-cell proliferation cytokine IL-2 and interferon (IFN)-α2b therapy[2]. Recent novel immunotherapies targeting immune checkpoints as standard of care have transformed the treatment paradigm of ccRCC[3,4]. However, a substantial subset of renal cancer patients do not respond to these therapies and patients who initially do respond eventually progress[5,6]. Cytotoxic tumor-infiltrating lymphocytes (TILs), in particular CD8+ T cells are key effectors of the adaptive antitumor immune response[7] and abundance of CD8+ T cells in solid cancers is generally associated with better survival in cancer patients[8–11]. However, in ccRCC, immune cell abundance is inversely correlated with survival, specifically TILs including CD8+ T cells[12–15]. Biomarker analysis results from recent clinical trials also supported the negative prognostic significance of T-cell infiltrates in the absence of immunotherapy within treatment-naive ccRCC patients[16,17]. Other abundant immune players in the ccRCC tumor microenvironment include monocytes, dendritic cells, and TAMs[18] that are now just starting to be studied.

Quantifying and inferring immune cell abundance from transcriptional analysis of bulk tumor samples is inadequate to provide a clear picture of the immune cell types[19,20]. While these studies are suggestive, they lack single-cell resolution for characterizing heterogeneous cell subpopulations that ultimately shape antitumor response, as has been demonstrated in breast cancer and melanoma[21,22]. Single-cell methodologies including flow cytometry, immunohistochemistry, and mass cytometry[14,18,23] have revealed immune cell states in ccRCC as discrete phenotypes when in vivo they typically display diverse spectrum of differentiation or activation states. Also, these methods require use of antibody panels targeting known immune cell components, and by design are not capable of identifying novel subpopulations of cells. SCRS has enabled comprehensive characterization of heterogeneous lymphoid and myeloid immune cells in several cancers[24–27], providing an unbiased approach to profiling cells and enabling molecular classification of different subpopulations and identification of novel gene programs. Transcriptomic mapping of T-lymphocytes coupled with TCR sequencing allows additional measurement of clonal T-cell response to cancer at an unprecedented depth[28,29].

Here, we report the single-cell profiling of the immune landscape in ccRCC mapping 25,688 of immune single cells (5′-sequencing and recombined V(D)J region of the T-cell receptor) in matched tumor samples and peripheral bloods from three treatment-naive ccRCC patients. We further integrated an additional, 11,367 immune cells isolated from peripheral blood and renal parenchyma[30] providing controls to evaluate tumor-specific transcriptional and clonal changes in immune populations at the single-cell level. Analysis of tumor-infiltrating T cells demonstrated distinct expression changes compared to peripheral blood and normal renal parenchyma. Clonal structure of T cells differed—with marked expansion seen in CD8+ T cells but not CD4+ T cells—and was associated with transcriptional patterning revealed by cell trajectory analysis. In myeloid cells, we observed an overall increase in macrophage populations with mixed polarization across patients. Predictive models derived from the CD8+ T cells and TAMs identified worse overall survival associated with proliferative CD8+ T cells and CD207+ TAMs. This represents the first such report of the immune landscape of ccRCC using SCRS for both transcriptional and clonal assessment.

## Results

### Single-cell expression profiling of immune cells in ccRCC. In order to define the immune microenvironment of human ccRCC,

we performed SCRS on flow-sorted lymphoid and myeloid cells from tumors and matched peripheral blood from three treatment-naive ccRCC patients. The general workflow for isolation and sequencing is available in Supplementary Fig. 1. To these samples, we added immune cells from peripheral blood and normal renal parenchyma[30] to obtain an integrated UMAP projection of 22 clusters across 37,055 primary immune cells (Fig. 1a). Across the three tissues, peripheral blood ($n = 21,160$), tumor ($n = 12,239$), and normal kidney ($n = 3556$), we found a number of clusters—notably clusters 0, 1, 3, and 7—sharing similar gene expression (Fig. 1b). Despite the integration of sequencing runs to reduce tissue-type divergence, each tissue type had enrichment for distinct clusters: peripheral blood formed the majority of Cluster 2, tumor tissues were enriched within Clusters 14, 17, 18, and 19, and normal kidney was enriched within Cluster 11. Based on gene expression, we assigned cell lineages to each cluster using a three-method approach: (1) expressions of canonical markers for T cells (CD3E, CD8A, CD4, and IL7R or CD127), B cells (CD19 and MS4A1), myeloid cells (CD14 and FCGR3A or CD16), and natural killer (NK) cells (KLRD1 and NKG7) (Fig. 1c), (2) correlations with gene signatures derived from purified cell populations deposited by ENCODE[31] (Fig. 1d), and (3) assignments of T-cell clonotypes based on the TCR sequencing. Based on these approaches we annotated clusters as monocytes (Clusters 0, 5, 11, 12, and 16), CD4+ T cells (Clusters 4, 6, 10, 13, 15, and 20), CD8+ T cells (Clusters 1, 8, 9, and 17), NK cells (Clusters 3 and 7), B cells (Cluster 2), macrophages (Cluster 14), and dendritic cells (DC, Clusters 18 and 19) (Fig. 1e). We also examined the relative proportion of cell types comprising the sequencing runs by tissue type (Fig. 1e). We observed a decrease of CD4+ T cells and B cells within normal kidneys or tumors relative to peripheral blood (Fig. 1e). Conversely and as expected, we also found an increase of CD8+ T cells and macrophages in tumors relative to peripheral bloods (Fig. 1e). Using high throughput immunohistochemistry on paired normal and tumor tissue, we found similar trends of increased CD8+ and decreased CD4+ T cells from tumor versus normal renal tissue derived from the ccRCC patient samples (Supplementary Fig. 2 and Supplementary Methods).

### Preferential overlap between peripheral blood and tumor CD8+ T lymphocytes. With the extensive literature demonstrating the role of TCR expansion in antitumor immunity and immunotherapy[11], we first wanted to investigate the dynamics of CD4+ and CD8+ T-cell clonal species in ccRCC. Using our previously described scRepertoire software[32], we assigned productive TCR sequences for TCRA and TCRB and defined clonotypes by the combination of both the genes and nucleotide sequences. For the identified T cells in ccRCC patients, recovering of at least one TCR chain, ranged from 74.8 to 87.6% after filtering and clonotype reconstruction. The complete table of clonotype information for the ccRCC samples is available in Supplementary Data 1. T-cell clonotypes had a clear distribution along the UMAP, with principal enrichment within Clusters 1, 4, 6, 8, 9, 13, 15, 17, and 20 (Fig. 2a). Cluster 21 was the exception for T cells, consisting of an estimated 19.6% of doublets and clustering with B cells, possibly indicating the cell–cell interaction of B and T cells and were excluded from further T-cell analyses. The frequency of clonotypes was assigned across patient samples, allowing for the quantification of clonotype numbers in the context of individual patients. We observed an increase in clonotype frequency principally in the CD8+ T-cell clusters (Fig. 2a). There were expanded clonotypes in assigned NK cell clusters 3 and 7, however, these clonotypes were also seen in other T-cell clusters, suggesting a possible subset of T cells with overlapping

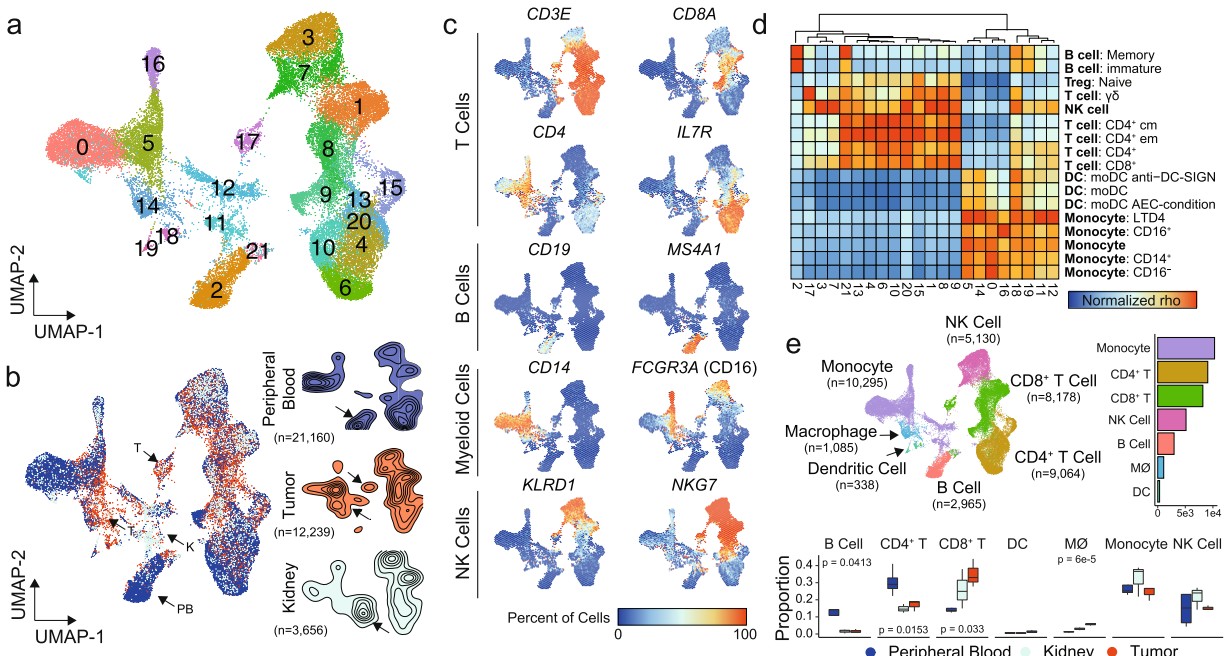

**Fig. 1 Single-cell RNA sequencing results from immune cells in ccRCC. a** UMAP of 37,055 primary immune cells of peripheral blood, normal renal parenchyma and tumor-infiltrating ccRCC patients. **b** Distribution of cells by tissue type, peripheral blood (blue), tumor (red), and kidney (light blue). Arrows indicated potential enriched or unique immune cells populations for tissue type. **c** Percent of cells expressing canonical immune cell markers across the UMAP. **d** Normalized correlation values for predicted immune cell phenotypes based on the SingleR R package for each cluster; dendrogram based on Euclidean distance. **e** UMAP demonstrating inferred immune cell types in ccRCC, clusters are colored by cell type and proportion of single-cell per sequencing run by tissue type. $P$ values based on one-way ANOVA; lack of labeled $p$ values equates to value >0.05.

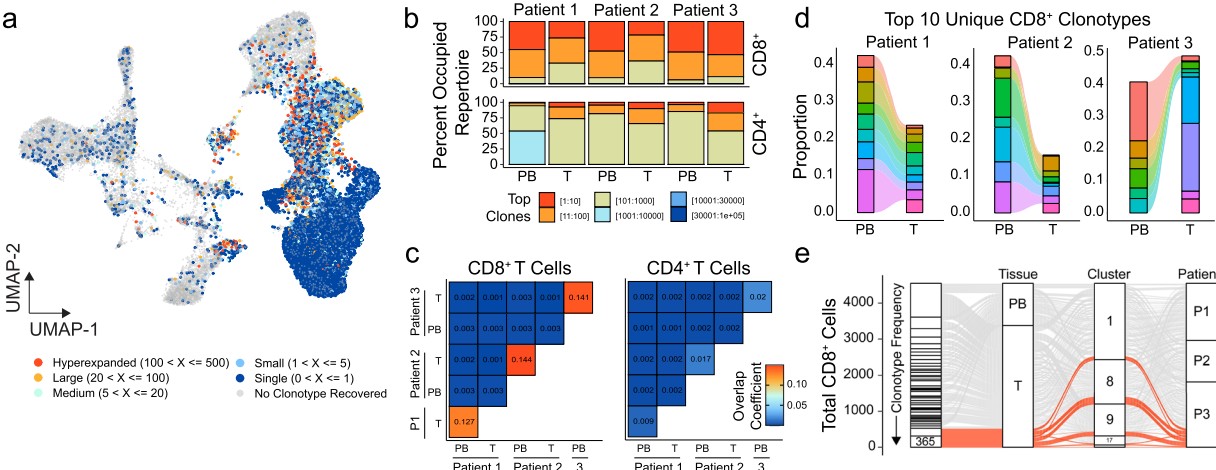

**Fig. 2 Clonal dynamics vary by T-cell types and patients. a** UMAP of 37,055 primary immune cells overlaid with the frequency of clonotypes assigned by sample identification. **b** Occupied repertoire space for the indicated clonotype groups for CD8+ and CD4+ T cells by sample and tissue type in ccRCC patients. **c** Clonal overlap quantification by overlap coefficient for CD8+ and CD4+ T cells by sample and tissue type in ccRCC patients. **d** The top ten clonotypes for each patient as a relative proportion of clonotypes for corresponding peripheral or tumor populations. Each color represents a unique clonotype by patient. **e** Distribution of clonotypes by tissue, UMAP cluster and ccRCC patient with highlighted (red) the top two clonotypes, comprising tumor-specific clonotypes across all clusters.

gene expression with NK cells or NK T cells. Single clones and clones with 1–5 copy numbers were seen across myeloid clusters (Fig. 2a), which may be a result of partial loss of finer gene expression differentiation during the expression integration[33]. Separating the T-cell classes, we noted a stark difference in clonotype space occupied by the top 10 clones in the CD8+ T cells compared to CD4+ T cells across ccRCC patients (Fig. 2b). This trend was consistent between the tumor-infiltrating and peripheral blood CD8+ T cells (Fig. 2b). We next asked if this

consistency in CD8+ T-cell expansion was a result of shared expanded clonotypes between tumor and peripheral blood. We found a relative patient-specific increase in shared clonotypes in CD8+ T cells compared to CD4+ T cells (Fig. 2c). We also noted that there was minimal overlap between patient clonotypes for both CD8+ and CD4+ T cells (Fig. 2c). The patient-specific overlap of CD8+ clonotypes showed relatively larger pools in peripheral-blood clonotypes contributing to the tumors (Fig. 2d). Interestingly, Patient 3—with the more advanced tumor stage

(pT3a compared to T1 of Patient 1 and 2)—showed expansion in tumor-specific clonotypes that was not seen in the blood (Fig. 2d). In the more advanced Patient 3 ccRCC, two clonotypes accounted for a total of 619 CD8+ T cells and were distributed across UMAP clusters (Fig. 2e), which supports the notion that T-cell clonotype is neither a determinant for UMAP clustering nor for functional indication. This compartmentalization of clonotypes associated with exhausted gene expression may reflect origin of the expansion in the tumor itself[34].

**CD8+ T cells in ccRCC tumors exhibit a transcriptional continuum with distinct populations.** Subclustering of CD8+ T cells revealed eight distinct clusters (Fig. 3a) with relative tissue-specific distribution (Fig. 3b). To understand the distribution of these new CD8+ subclusters along the UMAP, we first examined the relative percent of single cells represented in each cluster by tissue type. Tissue-infiltrating CD8+ T cells (both tumor and normal kidney) comprised the majority of CD8_0, CD8_1, CD8_3, CD8_5, CD8_6, and CD8_7. Only Clusters CD8_2 and CD8_4 had increased relative levels of peripheral-blood cells (Fig. 3b). Going from right to left across the x-axis of the UMAP, there is a change in tissue-specific contribution starting from peripheral blood (right) to kidney (middle) to increasing levels of ccRCC tumor CD8+ T cells (left), which may represent the

process of tissue infiltration itself. Within SCRS literature, there are concerns for variations in cell cycle leading to increased heterogeneity or obscure subpopulations[35], however proliferation of CD8+ T cells is an important surrogate marker of antitumor immune response[7]. We next examined the variation in proliferative gene signatures, finding a similar distribution to the tissue-type with increasing cells in S or G2M phases from right to left, peaking with Cluster CD8_6 (Fig. 3c).

In order to better characterize the CD8+ clusters, we used canonical and differential T cells markers to examine gene expression differences along the UMAP (Fig. 3d) with several patterns. The first pattern was the discovery of a naïve CCR7+ SELL+ TCF7+ being seen in CD8_4 (Fig. 3d). Looking for effector CD8+ T cells, we next observed two populations of IFNG+ PRF1+ T cells, principally in CD8_1 and CD8_0 (Fig. 3d). The latter also expressed immune checkpoints, such as CTLA4, HAVCR2, PDCD1, and TIGIT (Fig. 3d). These immune checkpoints were expressed at more moderate levels in both CD8_5 and CD8_6; however, CD8_6 exclusively expressed a number of proliferation markers, such as CDK1, MKI67, STMN1, and TOP2A (Fig. 3d). In order to examine gene expression patterns above single or selected genes, we used slingshot[36] to build minimum spanning trees between subclusters, generating curves based on the most varied genes (Fig. 3e). We identified five

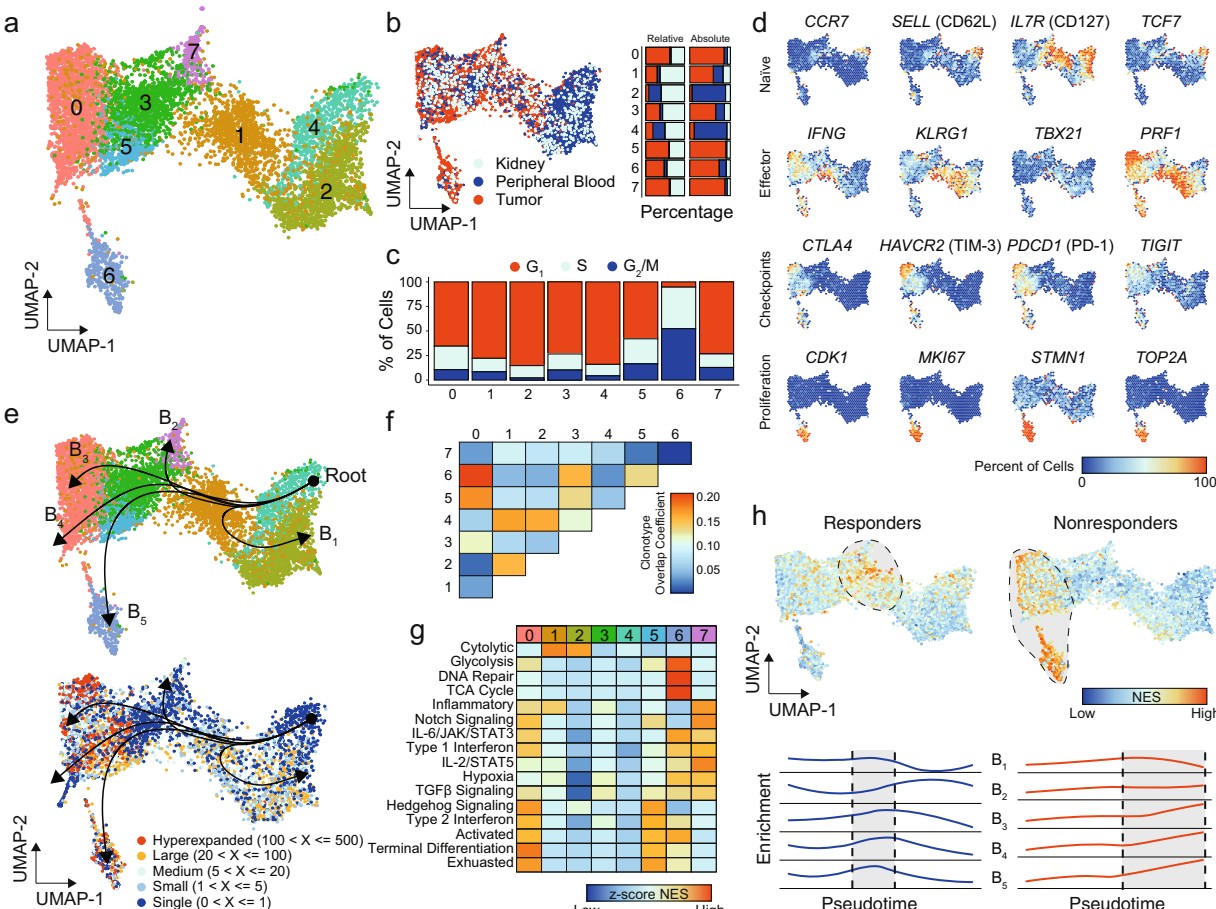

**Fig. 3 CD8+ T cells in ccRCC tumors exhibit a transcriptional continuum with distinct populations. a** UMAP subclustering of CD8+ T cells (original clusters 1, 8, 9, and 17). **b** UMAP distribution of single cells by tissue type with relative and absolute percent of cells by tissue in each cluster. **c** Cell-cycle regression assignments for CD8+ T cells by cluster assignment. **d** Percent of cells expressing selected markers for T-cell biology. **e** CD8+ UMAP of clusters (upper panel) and clonotype frequency (lower panel) overlaid with slingshot-based[36] cell trajectory starting at CD8_4 and proceeding into five distinct curves: branch 1 (B₁), B₂, B₃, B₄, and B₅. **f** Clonotype overlap coefficients between subclusters. **g** Z-transformed normalized enrichment scores from ssGSEA for selected gene sets by subcluster. **h** Normalized enrichment scores for therapeutic response or lack of response to anti-PD-1 therapy across the CD8+ T cells (upper panel) and by pseudotime of each branch (lower panel).

distinct curves (labeled $B_1$ to $B_5$) with the origin in the $CCR7^+$ $SELL^+$ $TCF7^+$ CD8_4. With the exception of B1 extending into CD8_2, the remaining curves graphed along a similar trajectory with a common node of CD8_1 and branching at distinct clusters with increased levels of tumor-infiltrating CD8$^+$ T cells (Fig. 3e). These curves also varied by CD8$^+$ T clonotypes based on TCR sequencing, with the root having no clonal expansion and the $B_3$, $B_4$, and $B_5$ curves terminating into regions with higher levels of clonal expansion compared to $B_1$ or $B_2$ (Fig. 3e). This clonotype relationship was also observed in specific clonotype sequences with overlapping clonotypes seen in subclusters CD8_0, CD8_6, CD8_5, and CD8_3 (Fig. 3f). In contrast, CD8_7 had minimal overlapping clonotypes with other subclusters (Fig. 3f). This relationship was seen also independent of the individual patient sequenced (Supplementary Fig. 3). In order to assess possible functional differences based on these branching, we performed gene set enrichment analysis (Fig. 3g)[37]. As expected based on the immune checkpoint inhibitors expression (Fig. 3d), Clusters CD8_0 and CD8_5 showed increased terminal differentiation and exhaustion (Fig. 3g). Cytolytic gene enrichment was seen in CD8_1, the $PRF1^+$ $IFNG^+$ population lacking immune checkpoints (Fig. 3g). The highly proliferative CD8_6 population was enriched for metabolic activity, such as the tricarboxylic acid cycle and glycolysis, and DNA repair (Fig. 3g). The $B_2$ curve termination cluster, CD8_7, has preferential enrichment of cytokine signaling, such as IL-2/STAT5, TGFβ, and type 1 IFN (Fig. 3g). With immune checkpoint inhibitor responsiveness associated with distinct CD8$^+$ T-cell populations[22], we next examined enrichment of signatures associated with response or nonresponse to anti-PD-1 therapies (Fig. 3h). Using the ordinal

construction of the trajectories, we created a pseudotime variable for cells, allowing us to see the difference in the enrichment along the curves. This approach found an overall enrichment in gene expression associated with responsiveness to anti-PD-1 at the terminal points of curve $B_2$ and midpoints of $B_3$, $B_4$, and $B_5$, corresponding to cells in CD8_1 (Fig. 3h, blue lines). Likewise, we observed an overall increase in gene expression associated with no response or progression on anti-PD-1 therapy at the terminal points of curves $B_3$, $B_4$, and $B_5$ (Fig. 3h, red lines).

**Single-cell CD4$^+$ T-cell characterization in ccRCC identifies disparate intratumoral populations.** CD4$^+$ T cells can influence cancer pathogenesis in various ways, either directly through cytolytic mechanisms or indirectly by modulating the tumor immune microenvironment. Subclustering of CD4$^+$ T cells revealed nine distinct clusters (Fig. 4a), with a similar pattern—as seen in CD8 T cells—of tissue distribution with predominantly peripheral-blood CD4$^+$ T cells on the right leading to tissue-infiltrating CD4$^+$ T cells on the left (Fig. 4b). The CD4_8 was composed solely of peripheral-blood cells from the healthy donor and was eliminated from the remaining analysis. Like the CD8$^+$ T cells, we next examined the canonical and differential T-cell markers along the UMAP (Fig. 4c). The first pattern that emerged was a naive $CCR7^+$ $SELL^+$ $TCF7^+$ being seen in CD4_1 and CD4_3 (Fig. 4c). Within the tumor-infiltrating CD4_4 cluster, we observed increased expression of the Th1 driver $TBX21$ (T-bet), activation marker $LAG3$ and $NR4A2$ and cytokine expression (Fig. 4c). Both CD4_5 and CD4_7 had expression of regulatory T (Tregs) cell markers (Fig. 4c), with higher levels of $FOXP3$,

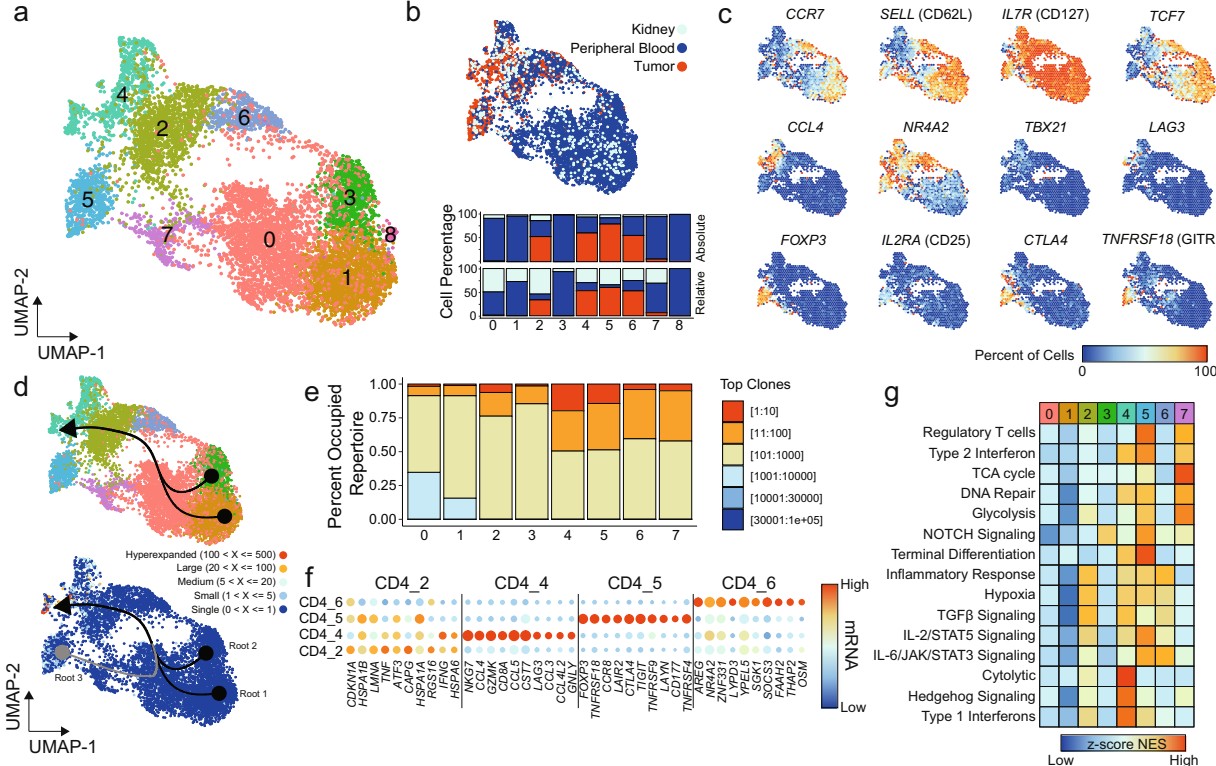

**Fig. 4 Single-cell CD4+ T-cell characterization within ccRCC. a** UMAP subclustering of CD4$^+$ T cells (original clusters 4, 6, 10, 13, 15, and 20). **b** UMAP distribution of single cells by tissue type with relative and absolute percent of cells by tissue in each cluster. **c** Percent of cells expressing selected markers for T-cell biology. **d** CD4$^+$ UMAP of subclusters (upper panel) and clonotype frequency (lower panel) overlaid with slingshot-based[36] cell trajectory starting from CD4_1 (root 1) and CD4_3 (root 2) with relative pseudotime for all curves calculated using slingshot. **e** Occupied repertoire space for CD4$^+$ subclusters. **f** Top ten markers for TI-predominant CD4$^+$ subclusters. Size of points are relative to percent of cells in the subcluster expressing the indicated mRNA species. **g** Z-transformed normalized enrichment scores from ssGSEA for selected gene sets by subcluster.

*IL2RA* (CD25), *CTLA4*, and *TNFRSF18* (GITR) in the tumor-predominant CD4_5 (Fig. 4c).

Constructing the cell trajectory curves based on the CD4+ subclustering, we observed two root points of the *CCR7+ SELL+ TCF7+* Clusters CD4_1 and CD4_3 leading to a common CD4_4 termination (Fig. 4d). Unlike the other CD4+ T cells, the curve generated for Tregs was divergent, starting at CD4_5 through CD4_7 and into CD4_4 (Fig. 4d). This likely represents a distinct expression pattern for Tregs (shared by CD4_5 and CD4_7) compared to other tumor-infiltrating CD4+ T cells. In addition, compared to the CD8+ subclustering, modest clonal expansion was seen in CD4_4 and CD4_5 and was not a clear pattern for cell trajectory (Fig. 4e). With the common termination point for the curves at Cluster CD4_4, we next wanted to examine if there are common markers for CD4+ T-cell infiltration in ccRCC by comparing tumor-infilterting to peripheral-blood CD4+ T cells. Within the tumor-infiltrating CD4+ T cell, 203 genes adjusted *p* value <0.05, log-fold change ≥0.5 and Δ cell percent >10% (Supplementary Data 2). Upregulated within the tumor-infiltrating CD4+ T cells were heat shock proteins (*HSPA1A* and *HSPA1B*), Jun and FOS constituents (*FOS*, *JUN*, and *JUNB*), MHC-II molecules (HLA-DRB), and secreted molecules (*CCL5*, *GZMA*, and *GZMK*) (Fig. 4f). Several of the upregulated genes are shared across all the tumor-predominant CD4 Clusters (Fig. 4f); however, each cluster also had unique expression markers. Both CD4_2 and CD4_4 had increased levels of *IFNG* (Fig. 4f), but CD4_2 was enriched for heat shock proteins, while CD4_4 had

a cytotoxic component and there was expression of *CD8A*, which likely represents modest contamination of CD8+ T cells (Fig. 4f). The tumor-infiltrating Tregs, CD4_5, had high levels of *CTLA4*, GITR (*TNFRSF18*) and *TIGIT*. In addition, CD4_5 had the highly-specific expression *CCR8* and *LAYN*, corresponding to previous reports[38,39]. The CD4_6 cluster had increased expression of the IL-6 cytokine, *OSM6*, and *AREG* and *SOCS3*, downstream of interleukin signaling (Fig. 4f). The differential expression closely matched the pathway analysis, with CD4_4 enriched for cytolytic and type I IFN signaling (Fig. 4g). The CD4_5 and CD4_7 Treg cluster had preferential enrichment for metabolic pathways, with high levels of terminal differentiation in tumor-infiltrated CD4_5 (Fig. 4g). The *OSM*high CD4_6 was enriched for IL-6/JAK/STAT3 signaling and inflammatory response genes (Fig. 4g).

**Prominent infiltrating macrophages in ccRCC have transcriptional divergence.** With the previous observation of an overall increase in macrophages and decreased monocytes in the integrated UMAP (Fig. 1e), we next focused on differential analyses of the myeloid populations (Fig. 5a). Across monocytes, macrophages and dendritic cells, subclustering found 20 distinct clusters (Fig. 5a). Tissue-specific distribution was observed, with the majority tumor-infiltrating myeloid cells in subclusters 0, 3, 7, 8, and 15 (Fig. 5b). In contrast, both normal kidney parenchyma and peripheral blood were comprised of a majority of monocytic

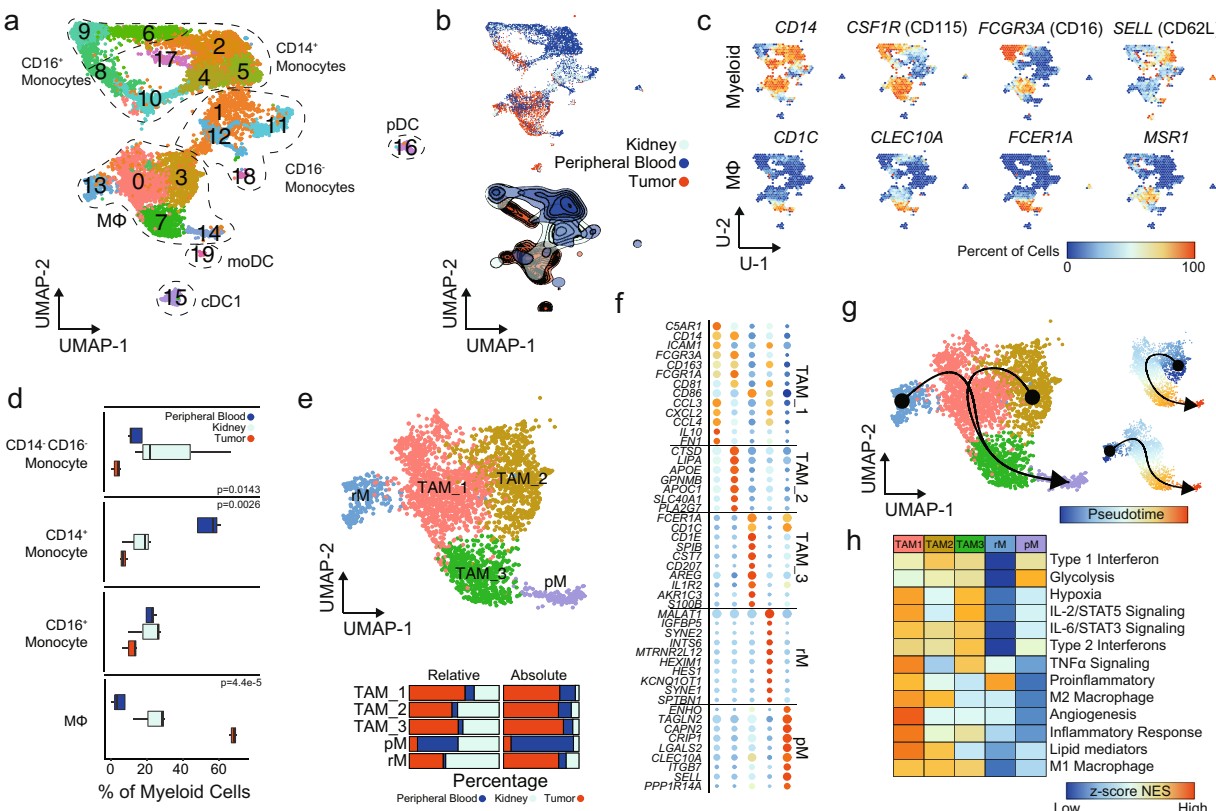

**Fig. 5 Single-cell myeloid characterization in ccRCC. a** UMAP subclustering of myeloid cells (original clusters 4, 6, 10, 13, 15, and 20). **b** UMAP distribution of single cells by tissue type with relative percent of cells by tissue in each cluster. **c** Percent of cells expressing selected markers for myeloid and macrophage markers. *P* values derived from one-way ANOVA testing. **d** Proportion of assigned cell types compared to total antigen presenting cells by tissue type. **e** Macrophage subclusters: tumor-associated macrophage 1 (TAM_1) (*n* = 1262), TAM_2 (*n* = 840), TAM_3 (*n* = 594), peripheral macrophage (pM) (*n* = 275), and resident macrophage (rM) (*n* = 194) with relative and absolute percent of cells by tissue in each cluster. **f** Top differential expression markers for macrophage subclusters. **g** Macrophage UMAP overlaid with slingshot-based[36] cell trajectories starting at rM and TAM_2 and proceeding into pM. Smaller UMAP shows pseudotime created by the cell trajectories. **h** Z-transformed normalized enrichment scores from ssGSEA for selected gene sets by subcluster.

subclusters (Fig. 5b). Populations were assigned using canonical markers and in addition to the previously described singleR approach with macrophage subclusters (0, 3, 7, 13, and 14) identified using markers such as *CD1C*, *CLEC10A* (CD301), *FCER1A*, and *MSR1* (Fig. 5c). In total, we observed a decreased proportion in CD14+ monocytes in tissue-infiltrating myeloid cells compared to peripheral blood and an increase in macrophages (Fig. 5d). Normal renal parenchyma had a variable increase in CD14− CD16− monocytes, which was not significant (Fig. 5d). As previously seen, we found a small number of DC (subclusters 15, 16, and 19) with distinct expression profiles associated with conventional DC1 (cDC1), plasmacytoid DC (pDC), and monocyte-derived DCs (moDC), respectively (Supplementary Fig. 4).

Next we isolated the five macrophage subclusters, relabeling them tumor-associated macrophage 1 (TAM_1), TAM_2, TAM_3, resident macrophage (rM), and peripheral macrophage (pM) based on the relative percent of cells derived from the respective tissue (Fig. 5e). Although similar in distribution along the UMAP, which preserves global structure of expression, these five clusters had distinct expression patterns (Figs. 1e, 5e). For example, the CD88high (*C5AR1*) CD54+ (*ICAM1*) TAM_1 expressed increased levels of chemokines and cytokines, like *CCL3*, *CCL4*, *CXCL2*, and *IL10*; CD64high (*FCGR1A*) CD16high (*FCGR3A*) TAM_2 subcluster expressed the apolipoprotein gene *APOE*, lysosomal lipase (*LIPA*), and ferroportin (*SLC40A1*); and CD1Chigh CD86+ TAM_3 had high levels of *IL1R2* and Langerhin (*CD207*), a marker of the skin-resident Langerhans cells (Fig. 5f). The pM subcluster had the highest level of the cell adhesion molecules *CLEC10A*, *SELL* (CD62L), and *ITGB7*, which

can dimerize with *ITGA4* (CD49d) or *ITGAE* (CD103). Like the CD8+ T cells, we built cell trajectories based on varied genes and found two distinct curves converging into TAM_3 and pM (Fig. 5g). In order to assess potential functional differences for the macrophages, we performed gene set enrichment analysis (Fig. 5h). As previously observed in single-cell data[25], no subclusters were distinctly M1 or M2 polarized. For example, TAM_1 had enrichment for gene sets commonly associated with the M2 macrophage compartment, such as angiogenesis and the production of lipid mediators, while also having the highest levels of TNFα signaling enrichment, a common M1 macrophage characteristic. Across the three TAM subclusters, modest enrichment in both type 1 and type 2 IFN signaling was observed (Fig. 5h). The non-TAM subclusters, rM and pM, had relatively lower levels of enrichment with the exception of proinflammatory signaling and glycolysis, respectively (Fig. 5h). We also found an increase in antigen processing and presentation of lipid antigens via MHC-I in TAM_2 and TAM_3, while TAM_1 had higher enrichment for polysaccharide antigens (Supplementary Fig. 5).

**Differential prognostic significance in CD8+ T cell and TAM subclusters.** These data demonstrate transcriptional differences in CD8+ T cells and TAMs in ccRCC. To determine if these transcriptional differences led to functional differences in tumor response, we investigated whether gene signatures can be developed from our SCRS data with prognostic values (Fig. 6a). Using the Cancer Genome Atlas data set for ccRCC[19], we separated the cohort in half, yielding a training and testing set. We isolated significantly upregulated genes from each subcluster of CD8+

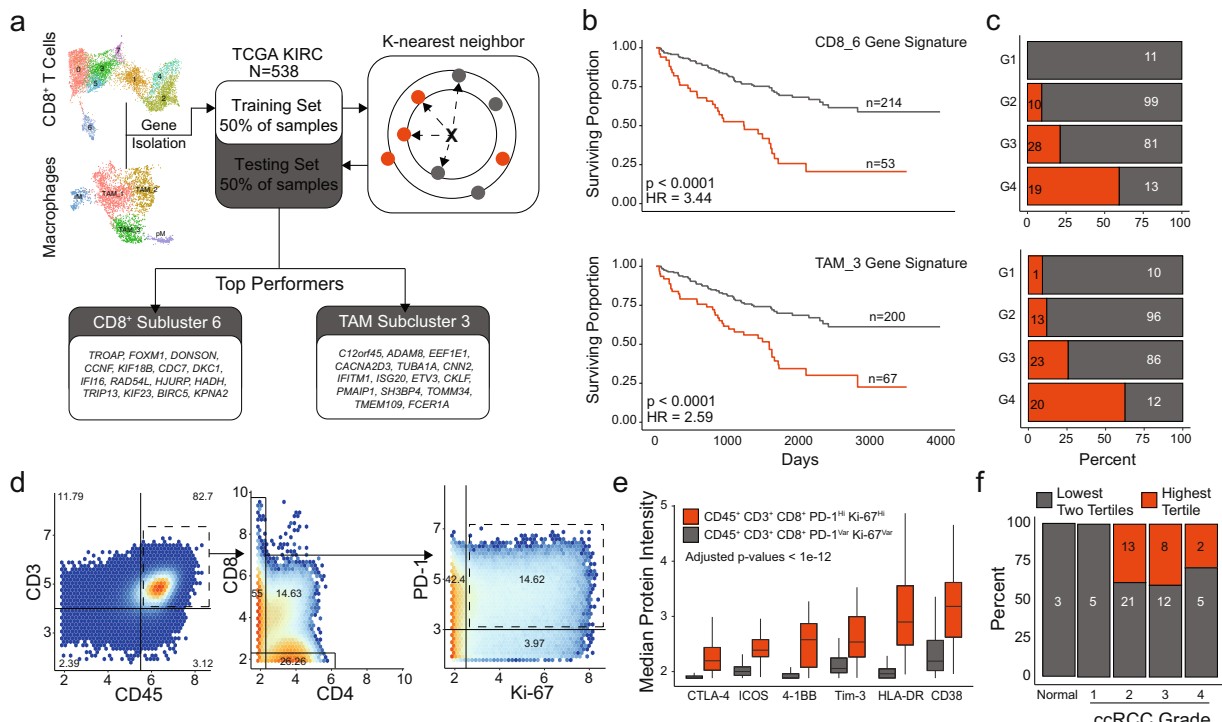

**Fig. 6 Prognostic values of gene signatures derived from CD8+ T cell and TAM subclusters. a** Schematic diagram of the machine-learning approach for signature development, selection and testing based on the k-nearest neighbors algorithm using the TCGA renal clear cell carcinoma data set. **b** Kaplan–Meier curves for overall survival in testing subset for the CD8_6 subcluster signature with corresponding distribution of histological grades by model assignment. **c** Kaplan–Meier curves for overall survival in testing subset for the TAM_3 subcluster signature with corresponding distribution of histological grades by model assignment with logrank *p* values. **d** Density plots for 937,713 T cells isolated from four healthy samples, 68 primary ccRCC, four metastasis, and quantified using mass cytometry[18]. **e** Markers with increased median expression in CD45+ CD3+ CD8+ PD-1+ Ki-67+ cells compared to other CD8+ cells. Adjusted *p* values < 1e−12 for all indicated markers using Welch's *T* test. **f** ccRCC primary tumor and healthy samples subdivided into tertiles by the proportion of CD45+ CD3+ CD8+ PD-1+ Ki-67+ relative to the entire CD45+ CD3+ CD8+ pool by histological grade.

T cells and macrophages selecting the top models for each cell type based on training results. Interestingly, we saw a consistently high performance for overall survival discrimination in CD8_6 and TAM_3-based signatures across all models we trained and different sizes of gene signatures. Applying the models to the testing cohort of 267 primary tumors, we found that both signatures had strong performance and categorized roughly 25% of ccRCC into poor prognostic groups and equating to hazard ratios of 3.44 and 2.59, respectively (Fig. 6b). We also observed that the poor-prognosis predictions were associated with increasing histological grades (Fig. 6c). There was not a clear association in expression by histological grade across genes in each signature (Supplementary Fig. 6). However, there was a significant association between the CD8_6 and TAM_3 classifications, which shared a high degree of overlap in patients classified into good-prognosis (188 in both signatures) and poor-prognosis (35 in both signatures), Fisher $p$ value = 9.3e−15. Interestingly despite this close association, the CD8_6 signature was more broadly applicable in discriminating overall survival across TCGA data sets (Supplementary Fig. 7).

Moving beyond mRNA expression, we wanted to see if we could identify the proliferative CD8_6 subcluster in tumor samples. Using mass cytometry data for T cells isolated from four healthy tissue samples, 68 ccRCC primary tumors, and four ccRCC metastasis, we identified a PD-1$^+$ Ki-67$^{Hi}$ subset in 14.6% of a CD45$^+$ CD3$^+$ CD8$^+$ T cells (Fig. 6d). The majority of CD45$^+$ CD3$^+$ CD8$^+$ T cells were either PD-1$^+$ Ki-67$^-$ (42.4%) or PD-1$^-$ Ki-67$^-$ (39%) (Fig. 6d). In addition to PD-1, this proliferative subset of CD8$^+$ T cells in ccRCC had increased levels of CTLA-4, ICOS, 4-1BB (CD137), TIM-3, HLA-DR, and CD38 compared to the other CD45$^+$ CD3$^+$ CD8$^+$ T cells (Fig. 6e). Calculating the proportion of PD-1$^+$ Ki-67$^{Hi}$ cells to total CD8$^+$ T cells by sample, we categorized samples into thirds. We observed a similar distribution to the CD8_6 gene signature assignments of the highest tertile for PD-1$^+$ Ki-67$^{Hi}$ cells in more advanced histological grades (Fig. 6f).

## Discussion

With the improved understanding on how immunotherapies work, the phenotypic and functional profile of immune cells in the tumor microenvironment is now well known to influence prognosis and disease outcome. Comprehensive knowledge of gene signatures to fully understand the roles of specific immune populations in cancer is of high pathological relevance; not only in identifying dysregulated immune determinants of cancer progression, but also as a useful tool for selecting patients, evaluating the likelihood of benefit from immunotherapy and further identifying clinically significant subpopulations.

Despite immunotherapy being a mainstay of treatment for advanced and treatment-naive ccRCC[3,40,41], ccRCC tumors have numerous counterintuitive immune findings compared with other immunotherapy-responsive tumors[42]. For example, unlike other tumors that respond to immune checkpoint blockade, ccRCC has a relatively low tumor mutational load, which is thought to drive T-cell infiltration[19,43], and mutational burden in ccRCC is not associated with response to anti-PD-1 therapy[15]. Moreover, despite this low mutational burden, ccRCC has the highest T-cell infiltration score among tumor types within the TCGA[44]. Similarly, therapeutic responses to anti-PD-1 therapy have been correlated with HLA heterozygosity in lung cancers and melanoma[45], which does not seem to be the case for ccRCC[15]. Taken together, these disparate findings suggest a more complex interrelationship of the immune compartment for ccRCC tumors. While a few recent studies have explored human ccRCC at a single-cell level[30,34,46,47], the technique has not yet

been applied to tumor-infiltrating immune cells to characterize the global transcriptional immune and T-cell receptor landscape along with underlying mechanisms contributing to this unique tumor environment.

T cells are recognized as key effectors of the adaptive antitumor immune response. Several studies have demonstrated association of these cells with an unfavorable response to therapy and poor patient survival in ccRCC[6,13]. In a comprehensive study, T cells represented the dominant lymphocytic population in most ccRCC cases and B cells were rarely detected[23], consistent with our findings of increased CD4$^+$ and CD8$^+$ T cells (Fig. 1). We found that the blood CD8$^+$ T cells are non-heterogeneous and poorly reflect tumor-infiltrating CD8$^+$ T-cell transcriptional profiles (Fig. 3a, d). Organizing the structure of the CD8$^+$ T-cell manifold, we found four distinct branches that may represent transcriptional states upon tumor infiltration, two associated with a PD-1$^+$ TIM-3$^+$ exhausted subcluster, a proliferative subcluster, and a fourth with the higher levels of cytokine signaling (Fig. 3e, g). The latter cluster, CD8_7 was also unique with minimal overlap in clonotypes compared to the other tumor-infiltrating predominant subcluster. Recent single-cell analyses in melanoma showed CD8$^+$ T cells with lower activation and exhausted expression patterns were associated with improved anti-PD-1 responses[22]. These responsive T cells had minimal shared clonotypes, similar to CD8_7[22]. Other studies have found the ccRCC tumors polyclonal CD8$^+$ T cells with an "immune-regulated" phenotype and lower cytotoxicity compared to tumors with oligoclonal CD8$^+$ T cells[48]. Recent SCRS studies of pre- versus post-treatment of anti-PD-1 in basal cell carcinoma have found increased number and clonal expansion of CD39$^+$ CD8$^+$ T cells after immunotherapy[49]. However, CD39$^+$ CD8$^+$ T cells in ccRCC have been shown to be associated with increased pathological stage and poor overall survival[50]. Based on gene expression, our CD8_0 and CD8_6 subclusters closely fit this population of cells and these clusters had 57% and 46.5% of cells from the advanced-stage Patient 3, respectively. In developing the CD8 signature, we found that the model discriminated overall survival, but also was associated with increasing histological grade, suggesting that more aggressive histological features are also correlated with a unique transcriptional response (Fig. 6b, c). Interestingly, patients with higher numbers of CD39$^+$ CD8$^+$ T cells had improved responses to sunitinib, a multi-tyrosine kinase inhibitor, suggesting that evaluation of exhausted phenotype for CD8$^+$ T cells may help in clinical decision making or therapy selection[50]. This is particularly interesting as we found shared CD8$^+$, but not CD4$^+$, T-cell clonotypes in the corresponding peripheral blood of ccRCC patients (Fig. 2). Although we find a stable overlap coefficient of around 13% for CD8$^+$ clonotypes, more work is needed to assess the dynamics of infiltration versus exfiltration on the CD8$^+$ T lymphocytes into the tumor bed.

The exhausted CD8$^+$ T-cell phenotype has been associated with advanced histological features and increased risk of disease progression[44,48,50], increased dysfunctional DC[6], and increased macrophage populations[18]. However, controversy surrounds the role of myeloid populations in ccRCC tumor prognosis and progression. This may, in part, be a result of transcriptional and phenotypic plasticity of tumor-infiltrating myeloid cells[18,25]. Our analysis demonstrated distinct CD16$^+$ myeloid population derived within tumor compared to peripheral blood or normal renal parenchyma and an overall increase in tumor-associated macrophages (Fig. 5a, d). M2 markers, like CD163 and CD204, have been associated with poor clinical outcomes in ccRCC[18,51] and were the highest in the TAM_1 and TAM_2 subclusters (Fig. 5g). This is despite no clear identification of canonical M1 or M2 macrophages subclusters (Fig. 5h). Model training for gene signatures for TAMs found better overall discrimination using

genes derived from TAM_3 (Fig. 6b), a subcluster that was unique for having lower levels of gene enrichment for M2 macrophages, angiogenesis, and lipid mediator production (Fig. 5h). The TAM_3 classification had an independently high degree with the CD8_6, suggesting the possible interaction or coordination between lymphoid and myeloid cells in ccRCC. The increased immunogenicity of ccRCC has been tied to upregulation of the antigen presenting machinery expression through MHC-I[44]. Across the myeloid subclustering, there was gene enrichment for MHC class I processing and presentation machinery in the DC subsets, while the macrophages had increased MHC class II enrichment (Supplementary Fig. 5). Although several distinct DC populations were detected and there was a trend for increase in cDC1 subset (subcluster 15) with known function of tumor antigen cross-presentation, further analysis was limited by the total number of DCs isolated.

Our strategy of single-cell analysis performed on immune cells taking into consideration the frequencies of lymphoid and myeloid cells during flow sorting provides a powerful way to identify the relationship between proportion of cell types and corresponding immune cell states. With the population structure and gene programs defined in our study for multiple immune populations, we showed that though the patients showed variable proportions of lymphoid and myeloid in each cell state, the number of states remain limited. Our study has limitations, including the small total number of samples ($n = 7$), the rarity of certain cell populations, and the need for further functional characterizations of these immune populations. However, taken together, we provide a transcriptional and clonotypic map of ccRCC immune cells that in hope to gain insight into biomarkers and therapeutic targets in ccRCC.

## Methods

**Subject details and tissue collection**. Paired blood and primary ccRCC along with matched normal kidney parenchyma samples were obtained from the University of Iowa Tissue Procurement Core and GUMER repository through the Holden Comprehensive Cancer Center from de-identified three subjects previously provided written consent approved by the University of Iowa Institutional Review Board (IRB) under the IRB number 201304826 and conducted under the Declaration of Helsinki Principles. The patients were males with an age range of 67–74 years old. Tumor grades were histologically determined by a pathologist. Primary tumor stages for Patient 1 and Patient 2 were reported as pT1b without extension, while Patient 3 was reported as pT3a with renal vein invasion.

**Tumor dissociation and isolation of mononuclear cells**. Renal tumor samples were dissociated into single cells by a semi-automated combined mechanical/enzymatic process. The tumor tissue was cut into pieces of (2–3 mm) in size and transferred to C Tubes (Miltenyi Biotec, Bergisch Gladbach, Germany) containing a mix of Enzymes H, R and A (Tumor Dissociation Kit, human; Miltenyi Biotec). Mechanical dissociation was accomplished by performing three consecutive automated steps on the gentleMACS™ Dissociator (h_tumor_01, h_tumor_02, and h_tumor_03). To allow for enzymatic digestion, the C tube was rotated continuously for 30 min at 37 °C, after the first and second mechanical dissociation step[52]. Cells from fresh tumor specimens were incubated with FcR blocking reagent (StemCell Technologies, Vancouver, Canada) for 10 min at 4 °C and labeled with 1 µg/ml of the FITC anti-human CD45 antibody (BioLegend, San Diego, CA) per $10^7$ cells for 20 min at 4 °C. CD45$^+$ cells were isolated using the EasySep™ FITC Positive Selection Kit (StemCell Technologies). Alternatively, mononuclear cells from whole peripheral blood of paired subjects were isolated using SepMate Tubes (StemCell Technologies) by density gradient centrifugation. Cells were then viably frozen in 5% DMSO in RPMI complemented with 95% FBS. Cryopreserved cells were resuscitated for flow cytometry analyses by rapid thawing and slow dilution.

**Cell sorting for single-cell RNA sequencing**. Viable immune (CD45$^+$ Hoechst$^−$) single-cell suspensions generated from three ccRCC tumor samples and blood were FACS sorted on a FACS ARIA sorter (BD Biosciences) for lymphoid and myeloid cells (ratio 3:1). This was to consistent sequencing depth for both myeloid and lymphoid cells across the three patients, as myeloid cells have 3–10-fold greater feature expression. The cells were sorted into ice-cold Dulbecco's PBS + 0.04% non-acetylated BSA (New England BioLabs, Ipswich, MA). Sorted cells were then

counted and assessed viability MoxiGoII counter (Orflo Technologies, Ketchum, ID) ensuring that cells were resuspended at 1000 cells/µl with a viability >90%.

**Library preparation, single-cell 5′, and TCR sequencing**. Single-cell library preparation was carried out as per the 10× Genomics Chromium Single-Cell 5′ Library and Gel Bead Kit v2 #1000014 (10× Genomics, Pleasanton, CA). Cell suspensions were loaded onto a Chromium Single-Cell Chip along with the reverse transcription (RT) master mix and single-cell 5′ gel beads, aiming for 7500 cells per channel. Following generation of single-cell gel bead-in-emulsions (GEMs), RT was performed using a C1000 Touch Thermal Cycler (Bio-Rad Laboratories, Hercules, CA); 13 cycles were used for cDNA amplification. Amplified cDNA was purified using SPRIselect beads (Beckman Coulter, Lane Cove, NSW, Australia) as per the manufacturer's recommended parameters. Post-cDNA amplification reaction QC and quantification was performed on the Agilent 2100 Bioanalyzer using the DNA High Sensitivity chip. For input into the gene expression library construction, 50 ng cDNA and 14 cycles was used. To obtain TCR repertoire profile, VDJ enrichment was carried out as per the Chromium Single Cell V(D)J Enrichment Kit, Human T cell #1000005 (10× Genomics) using the same input samples. Sequencing libraries were generated with unique sample indices for each sample and pooled. Libraries were sequenced on an Illumina HiSeq 4000 using a 150-pair-end sequencing kit. Gene expression FASTQ files were aligned to the human genome (GRCh38) using the Cell Ranger v2.2 pipeline, while clonotype sequencing was aligned to the vdj_GRCh38_alts_ensembl genome build provided by the manufacturer.

**Incorporation of other SCRS data sets**. SCRS and TCR sequencing data processed using Cell Ranger v2.2 for healthy donor peripheral-blood immune cells were acquired from the 10× Genomics website on 6/20/2020. Filtered gene matrix and contig annotations were used in the incorporation of the uniform manifold approximation and project (UMAP). Total number of cells from healthy peripheral-blood control were 7726. SCRS of normal immune populations in the kidney were derived from previously published data[30]. Gene expression matrices were downloaded from the EGAS00001002325 and filtered for normal renal parenchyma cells using the provided cell manifest for the samples RCC1, RCC2, and RCC3. These samples were processed using the procedure as described below to form a UMAP. Immune cells were identified using canonical markers for lineage and were then isolated. Isolated immune cells for normal renal parenchyma were: RCC1 ($n = 1011$), RCC2 ($n = 888$), and RCC3 ($n = 1757$).

**SCRS integration**. Initial processing of cells isolated from ccRCC patients; Patient 1 ($n = 10,694$), Patient 2 ($n = 5174$), and Patient 3 ($n = 9805$) were processed and integrated with the above samples using the Seurat R package (v3.0.2)[53,54]. We removed cells with a percentage of mitochondrial genes >15% and UMI > 5000 to control for multiplets. Samples were normalized using the *SCTtransform* approach[55] with default settings. Preparation for integration used 3000 anchor features and *PrepSCTIntegration*. The integration of sequencing runs occurred with the SCT-transformed data. The dimensional reduction to form the UMAP utilized the top 30 calculated dimensions and a resolution of 0.7. Data characteristics by sequencing run can be found in Supplementary Data 3. Cell-type subclustering used the SCTtransform approach as described for the whole-cell integration, but by integrating the data across samples instead of individual sequencing runs. The adjusted dimensional inputs for the subclustering analysis can be found in Supplementary Data 4. Parameters for UMAP generation and clustering were looped from across a range of 5–50 for dimensional inputs and 0.3–1.5 for resolution, final parameters were selected to generate consistent visualizations. Integration across the samples for subclustered populations is available in Supplementary Fig. 1. Doublet density estimation was performed across each cell using the scDblFinder (v1.4.0) R package using the top 30 PCA dimension and a $K$ of 50. Density scores of $\log 2(x + 1) \geq 3$ were designated as doublets in the integrated object and subcluster-based analyses (Supplementary Fig. 8).

**SCRS data analysis and visualizations**. The schex R package (v1.1.5) was used to visualize mRNA expression of lineage-specific or highly differential markers by converting the UMAP embedding into hexbin quantifications of the proportion of single cells with the indicated gene expressed. Default bins across all cells was 80 and 40 for subcluster analyses, unless otherwise indicated in the figure legend. This was done to prevent bias in expression evaluation generated by overlapping dot plots. Differential gene expression utilized the Wilcoxon rank sum test on count-level mRNA data. For differential gene expression across clusters or subclusters, *FindAllMarkers* function in the Seurat package using the log-fold change threshold >0.25, minimum group percentage = 10%, and the pseudocount = 0.1. Differential comparisons between conditions utilized the *FindMarkers* function in Seurat, without filtering and a pseudocount = 0.1. Multiple hypothesis correction was reported using the Bonferroni method. Cell-cycle assignment was performed in Seurat using the *CellCycleScoring* function and genes derived from Nestorowa et al.[56]. Genes were isolated by calling *cc.genes.updated.*2019 in Seurat.

Cell-type identification utilized the SingleR (v1.0.1) R package[57] with correlations of the single-cell expression values with transcriptional profiles from

pure cell populations in the ENCODE[31]. In addition to correlations, canonical markers for cell lineages (Supplementary Data 5) and corresponding TCR sequences were used. Gene set enrichment analysis was performed using the escape R package (v0.99.0). Gene sets were derived from the Hallmark library of the Molecular Signature Database and from previous publications[22,25]. Enrichment for anti-PD-1 therapy response was derived from Sade-Feldman et al. to develop gene signatures for the CD8_B (nonresponsive) and CD8_G (responsive) single-cell populations[22]. TCR analysis utilized our previously described scRepertoire R package (v0.99.3)[32] with clonotype being defined as the combination of the gene components of the VDJ and the nucleotide sequence for both TCRA and TCRB chains and assigned on the integrated Seurat object. Cell trajectory analysis used the slingshot (v1.6.0) R package[36] with default settings for the *slingshot* function and using the embedding from the subclustering for each cell type. Ranked importance of genes were calculated using the top 300 variable genes and rsample (v0.0.9) and tidymodels (v0.1.0) R packages were used to generate random forest models based on a training data set of 75% of the cells. The *rand_forest* function in the parsnip (v0.1.1) R package was used, with mtry set to 200, trees to 1400, and minimum number of data points in a node equal to 15 across all cell types. The processed data and code for all analyses will be made public upon publication at https://github.com/ncborcherding/ccRCC.

**Mass cytometry analysis**. Flow cytometry standard files were downloaded for 78 samples utilizing a previously-defined T-cell and tumor-associated macrophage panel[18]. Subsequent loading and analyses of the data were based on the accompanying published methods[18]. These files were loaded into R using the flowCore (v2.0.1) R package. Protein signal was arcsinh transformed using a cofactor of 5, filtered for previously identified T or myeloid cells. Further data visualization utilized ggplot2 (v3.3.1).

**Machine-learning modeling**. The renal clear cell carcinoma (KIRC) log2 gene expression data were downloaded from the University of California Santa Cruz Xena Browser and filtered for only primary tumor samples. Updated clinical information was assigned to the expression data using the tumor barcode[58]. Gene signatures from subcluster analysis were generated by comparing gene expression between clusters and filtering differential gene expression results for genes with >0.5 log-fold change and 15% difference in cell expression. Training and testing sample cohorts were divided using the *sample* function with *set.seed* set to 10, splitting the data into a 1:1 ratio. Feature selection was performed using recursive feature selection using the cross-validation method to optimize feature selection for 5, 10, 15, and 20 features in the caret (v6.0-86) R package. For each gene set, several models were trained, including support vector machines, bagged trees, and k-nearest neighbors. Authors selected the final models based on the performance of the trained results, k-nearest neighbor models had similar performance to the support vector machines, with the added benefit of classifying samples based on the nearest point to the training set in the selected feature space or classifying data points based on similarity. Selected models were then used to predict survival in the testing cohort and testing parameters were then calculated. Application of the PANCAN signature analysis was performed as described above using randomly selected 50% of the KIRC TCGA cohort for training and applying the KNN model across all samples with both RNA and overall survival data ($n = 11,014$) in the PANCAN batch-corrected RNA cohort. The testing was then separated by cancer type and Cox hazard ratio and logrank $p$ value were visualized. Survival analyses utilized the survival (3.1–12) and survMiner (v0.4.7) R packages.

**Statistics and reproducibility**. Statistical analyses were performed in R (v4.0.1). Two-sample significance testing utilized Welch's $T$ test, with significance testing for more than three samples utilizing one-way analysis of variance. Boxplots display 1.5 times the interquartile range unless otherwise indicated.

**Reporting summary**. Further information on research design is available in the Nature Research Reporting Summary linked to this article.

## Data availability
Quantified gene expression counts and V(D)J T-cell receptor sequences for single-cell RNA sequencing are available at the Gene Expression Omnibus at GSE121638. Any other data relevant to this study are available from the authors upon reasonable request.

## Code availability
The processed data and code for all analyses will be made public upon publication at https://github.com/ncborcherding/ccRCC. This data has also been deposited using Zenodo[59] under https://doi.org/10.5281/zenodo.4311825.

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

## Acknowledgements

Funding for this project was provided by Rock "N" Ride Foundation (PI: Y.Z.) and from the National Institute of Health under the R01 CA200673 (PI: W.Z.), R01 CA203834 (PI: W.Z.), K08 CA226391 (PI: R.W.J.), and F30 CA206255 (PI: N.B.). W.Z. was also supported by DOD/CDMRP grant BC180227 (PI: W.Z.), and an endowment from the Dr. and Mrs. James Robert Spencer Family Cancer Research Fund (PI: W.Z.) The flow cytometry and sequencing facilities are funded in part, by the National Cancer Institute of the National Institutes of Health under Award Number P30CA086862. The FACSAria Fusion high-speed cell sorter was supported with funds from the National Center for Research Resources of the National Institutes of Health under Award Number 1 S10 OD016199-01A1. The content is solely the responsibility of the authors and does not necessarily represent the official views of the National Institutes of Health. We thank Michael Knudson, Rita Sigmund, Joe Galbraith, Janice Cook-Granroth, Bethany Kilburg, and Celeste Charchalac from University of Iowa Carver College of Medicine, Tissue Procurement Core (TPC) and Genito-Urologic Tissue Repository (GUMER) for receiving biological samples and clinical data. We thank Justin Fishbaugh, Heath Vignes, and Michael Shey from the University of Iowa Flow Cytometry Facility. We thank Kevin Knudtson, Mary Boes, Garry Hauser, and Mari Eyestone from the Iowa Institute of Human Genetics (IIHG) Genomics Division for planning and assisting use of Next Gen Sequencing (NGS) platforms, Diana Kolb from the IIHG Bioinformatics Division and the University of Iowa High Performance Computing (HPC) facility.

## Author contributions

Conception and design: A.V., Y.Z., and W.Z. Development of methodology: N.B., A.P.V., W.Z., A.B., and J.K. Acquisition of data: K.N., Y.Z., A.V., A.B., and J.K. Analysis and interpretation of data: N.B., A.P.V., A.S., R.W.J., W.Z., and Y.Z. Writing, review, and/or revision of the paper: N.B., A.V., A.P.V., A.S., R.W.J., W.Z., and Y.Z. Supervision: Y.Z., W.Z., and R.W.J.

## Competing interests

R.W.J. has a financial interest in XSphera Biosciences Inc., a company focused on using ex vivo profiling technology to deliver functional, precision immune-oncology solutions for patients, providers, and drug development companies. R.W.J. interests were reviewed and are managed by Massachusetts General Hospital and Partners HealthCare in accordance with their conflict of interest policies. Y.Z. is on the advisory board of Amgen, Roche Diagnostics, Novartis, Janssen, Eisai, Exelixis, Castle Bioscience, Array, Bayer, Pfizer, Clovis, and EMD Serono. Y.Z. has received institutional clinical trial support from NewLink Genetics, Pfizer, Exelixis, and Eisai. These associations are not related to the work herein described in the paper. Other authors declare no competing interests.
