## [Peer Review File · Communications Biology]

Reviewers' comments:

Reviewer #1 (Remarks to the Author):

In this manuscript, authors analyzed expression profiles of flow sorted lymphoid and myeloid single cells of tumor and peripheral blood from three treatment-naïve ccRCC patients using single cell RNA-seq (scRNA-seq) technique. Based on cell clustering method, cell types were identified and characterized to identify sub-populations of immune cells. Each sub-cluster were described extensively with the gene expression profile. Finally authors proposed that the immune cells in tumor microenvironment of clear cell renal cell carcinoma (ccRCC) might be important for the response of immunotherapy on ccRCC. Although this paper covers immune cell profiles of ccRCC, supporting evidence on their conclusion was not fully demonstrated.

1. Three ccRCC patients seems to be not enough to generalize their findings as a immune landscape of ccRCC. At least, authors need to validate the pattern of sub-population of immune cells in another pool of patients with immunohistochemistry or other method.
2. Authors need to check their findings described as a specific to tumor in matched normal tissue immune cells. Immune profile in tissue and peripheral blood should be different even in healthy normal people.
3. Authors also need to compare healthy normal PBMC for their conclusion on ccRCC-specific feature of peripheral immune of patients.
4. What was unique immune feature of ccRCC? Is it possible to compare the distribution of immune sub-population with other cancer types from public dataset?

Reviewer #2 (Remarks to the Author):

Vishwakarma et al use single-cell RNA and TCR sequencing to survey the immune populations present in the tumor and peripheral blood of patients with clear cell renal cell carcinoma (ccRCC). They find that CD4+ and CD8+ T cell subpopulations are more heterogeneous in tumor compared to blood. Furthermore, the exhausted CD8+ T cell population was the most abundant in all three patients and was found exclusively in the tumor but not peripheral blood. TCR analysis revealed a substantial amount of overlap of TCRs between T cell subpopulations and between tumor and blood. The authors also describe the heterogeneity of the myeloid population within the tumor and blood, identifying both anti- and pro-inflammatory macrophage subsets, classical and non-classical monocytes, as well as monocyte-derived DCs. Overall, this is an important paper, which will be of great interest to the field and serve as a useful dataset for many future studies. The analyses were performed well and described thoroughly in the methods to enable reproducibility. We have a few suggestions that may improve the manuscript.

Major comments:

1. The pseudotime trajectory analysis of CD8+ T cell differentiation (Fig 2E) is interesting. One question that follows is the characterization of the cells around the branch point between exhausted and proliferative cells. Looking at the colors by eye, there appears to be an enrichment of cluster 10 cells at the branch point, although cells from this cluster are also spread along the projection from naïve to the branch point. It would be helpful to include a visualization of each cluster separately to show where cells from each cluster lie along the trajectory. Furthermore, it would be informative to incorporate the finer effector/memory T cell clusters (e1, e2, cm, rm1-3) into this pseudotime analysis. This would allow for more specific interpretation of which factors might be important in specifying T cell fate, which could be a step closer towards a more mechanistic understanding of T cell exhaustion in ccRCC.
2. The authors state that "enriched TCRs were more common in intratumoral CD8+ T cells compared to single TCRs in peripheral blood." However, in Fig. S5A, the difference between % CD8+ TCRs expanded in tumor and peripheral blood does not appear to be significant. Perhaps the authors could clarify this point.
3. It would be exciting to do a bit more with the TCR analysis, specifically in the integration with gene

expression data and phenotypic clusters. Some questions that could be answered using the data include:

- a. For the top 5 clonotypes shared in the tumor and peripheral blood (Fig 4C), what are the associated phenotypes of these clones? Is there enrichment of a certain CD4+ or CD8+ subpopulation among these shared clonotypes?
- b. Which CD4+ and CD8+ sub-populations are most clonally expanded? Is this consistent across patients? How might the level of expansion correspond to specific gene signatures?

Minor comments:

1. Comparing the tSNE colored by patients (Fig 1A, bottom right) and the t-SNE split by tumor and blood (Fig 1E), it seems the pink tumor CD8 T cell (T) cluster 0 is dominated by cells from patient 2. And from Table S1, cells from patient 2 make up >60% of total cells in this cluster. Is there some clinical/pathological indication for this exhaustion cluster would not be seen in patients 1 and 3? Perhaps some discussion on this subject would be interesting to the reader.
2. A slightly more detailed explanation of why lymphoid and myeloid cells were mixed in different proportions (7:3, as stated in Results section; 3:1 in Methods) would be helpful to the reader.
3. The examination of the exhausted CD8+ T cell population found in the tumor was informative, revealing sub-populations with distinct profiles of exhaustion markers (Fig 2D). However, the authors' claim that these subsets have 'distinct functional properties' might be overstated, since no functional experiments were performed to validate this statement.
4. Please provide a color legend for 4D.

Joy Pai and Ansu Satpathy

Reviewer #3 (Remarks to the Author):

Vishwakarma et al analyzed 24,904 lymphoid and myeloid cells in matched tumor and blood from 3 patients with clear cell renal cell carcinoma (ccRCC) by single-cell RNA sequencing using 10x genomics platform. Both single-cell transcriptome and lymphocyte repertoire were analyzed. This is a descriptive report aimed to characterize ccRCC immune landscape. This is considered as the first report of the immune landscape of ccRCC using scRNA-seq. However, none of the major conclusions were validated by a secondary method.

Another major flaw of this study is the authors ignored the multiplet issue by drop-let based approach. "Cell suspensions were loaded onto a Chromium Single-Cell Chip along with the reverse transcription (RT) master mix and single cell 5' gel beads, aiming for 7,500 cells per channel." at this recovery rate, approximately 6% of the cells are multiplets and the authors did not attempt to address how this would impact their analyses.

Reviewer #1 (Remarks to the Author):

In this manuscript, authors analyzed expression profiles of flow sorted lymphoid and myeloid single cells of tumor and peripheral blood from three treatment-naïve ccRCC patients using single cell RNA-seq (scRNA-seq) technique. Based on cell clustering method, cell types were identified and characterized to identify sub-populations of immune cells. Each sub-cluster were described extensively with the gene expression profile. Finally, authors proposed that the immune cells in tumor microenvironment of clear cell renal cell carcinoma (ccRCC) might be important for the response of immunotherapy on ccRCC. Although this paper covers immune cell profiles of ccRCC, supporting evidence on their conclusion was not fully demonstrated.

1. Three ccRCC patients seems to be not enough to generalize their findings as an immune landscape of ccRCC. At least, authors need to validate the pattern of sub-population of immune cells in another pool of patients with immunohistochemistry or other method.

Response: We have added additional analysis and data sources. After identifying proliferative CD8+ T cells as a poor prognostic marker, we used mass CyTOF data from Sade-Feldman et alia 2018 to identify the cell population across 68 patients with enrichment in advanced-grade diseases (Figure 6).

2. Authors need to check their findings described as a specific to tumor in matched normal tissue immune cells. Immune profile in tissue and peripheral blood should be different even in healthy normal people.

3. Authors also need to compare healthy normal PBMC for their conclusion on ccRCC-specific feature of peripheral immune of patients.

Responses to 2-3: We have added scRNA-sequencing immune cell results from a healthy peripheral blood and 3 normal kidney tissues (a total of 11,382 more cells). At a global level, we observed an increase in unique myeloid cells in normal renal tissue (Figure 1B) and tumors were enriched for populations with expression consistent with CD8+ T cells and macrophages (Figure 1B). We did not observe a notable difference in healthy vs ccRCC peripheral blood samples at the global level. The overlap of all the samples can be visualized in Supplemental Figure 1.

4. What was unique immune feature of ccRCC? Is it possible to compare the distribution of immune sub-population with other cancer types from public dataset?

Response: This is an excellent question, especially as ccRCC seems to be unique in the immunotherapy-responsive solid tumors. Although beyond the scope of this specific project, we and others are looking into just that. We hope our publicly provided data will allow for additional analysis on this subject.

Reviewer #2 (Remarks to the Author):

Vishwakarma et al use single-cell RNA and TCR sequencing to survey the immune populations present in the tumor and peripheral blood of patients with clear cell renal cell carcinoma (ccRCC). They find that CD4+ and CD8+ T cell subpopulations are more heterogeneous in tumor compared to blood. Furthermore, the exhausted CD8+ T cell population was the most abundant in all three patients and was found exclusively in the tumor but not peripheral blood. TCR analysis revealed a substantial amount of overlap of TCRs between T cell subpopulations and between tumor and blood. The authors also describe the heterogeneity of the myeloid population within the tumor and blood, identifying both anti- and pro-inflammatory macrophage subsets, classical and non-classical monocytes, as well as monocyte-derived DCs. Overall, this is an important paper, which will be of great interest to the field and serve as a useful dataset for many future studies. The analyses were performed well and described thoroughly in the methods to enable reproducibility. We have a few suggestions that may improve the manuscript.

Major comments:

1. The pseudotime trajectory analysis of CD8+ T cell differentiation (Fig 2E) is interesting. One question that follows is the characterization of the cells around the branch point between exhausted and proliferative cells. Looking at the colors by eye, there appears to be an enrichment of cluster 10 cells at the branch point, although cells from this cluster are also spread along the projection from naïve to the branch point. It would be helpful to include a visualization of each cluster separately to show where cells from each cluster lie along the trajectory. Furthermore, it would be informative to incorporate the finer effector/memory T cell clusters (e1, e2, cm, rm1-3) into this pseudotime analysis. This would allow for more specific interpretation of which factors might be important in specifying T cell fate, which could be a step closer towards a more mechanistic understanding of T cell exhaustion in ccRCC.

Response: As part of the suggestions by other reviewers, we have added additional samples and re-performed the trajectory analysis (Figure 3E), however, we do find similar divergence between exhausted and proliferative CD8+ T cells. We have changed the visualization to overlay onto the UMAP plot itself, to increase ease for readers.

2. The authors state that “enriched TCRs were more common in intratumoral CD8+ T cells compared to single TCRs in peripheral blood.” However, in Fig. S5A, the difference between % CD8+ TCRs expanded in tumor and peripheral blood does not appear to be significant. Perhaps the authors could clarify this point.

Response: We have substantially modified the clonotype analysis for both CD4+ and CD8+ T cells (see figure 2) and the discussion of clonotype analysis in the manuscript. We see a relatively stable clonotype overlap between the tumor and peripheral blood of the 3 patients (overlap coefficient 0.127-0.144) in CD8+ T cells, but not CD4+ T cells (with clonal expansion or overlap).

3. It would be exciting to do a bit more with the TCR analysis, specifically in the integration with gene expression data and phenotypic clusters. Some questions that could be answered using the data include:

a. For the top 5 clonotypes shared in the tumor and peripheral blood (Fig 4C), what are

the associated phenotypes of these clones? Is there enrichment of a certain CD4+ or CD8+ subpopulation among these shared clonotypes?

Response: As mentioned above, we have substantially modified the clonotype analysis as part of the revision process. We found, in general, the top clonotypes in patient 1 and 2 shared between tumor and blood (Figure 2D), while the top clonotypes in patient 3 were seen in tumor-specific manner.

b. Which CD4+ and CD8+ sub-populations are most clonally expanded? Is this consistent across patients? How might the level of expansion correspond to specific gene signatures?

Response: We have added analysis of clonotypes to sub-analysis of CD8+ T cells (Figure 3E,F) and CD4+ T cells (Figure 4D) and expanded the analyses to include discussion of gene signatures.

Minor comments:

1. Comparing the tSNE colored by patients (Fig 1A, bottom right) and the t-SNE split by tumor and blood (Fig 1E), it seems the pink tumor CD8 T cell (T) cluster 0 is dominated by cells from patient 2. And from Table S1, cells from patient 2 make up >60% of total cells in this cluster. Is there some clinical/pathological indication for this exhaustion cluster would not be seen in patients 1 and 3? Perhaps some discussion on this subject would be interesting to the reader.

Response: Although we have reclustered and renamed the patient samples (now patient 3), the reviewer makes a great point. As part of the expansion of clonotype and broader analysis, we note that this patient has 2 dominant clones in the tumor that accounts for tumor-exclusive populations. We have added discussion of this to the manuscript.

2. A slightly more detailed explanation of why lymphoid and myeloid cells were mixed in different proportions (7:3, as stated in Results section; 3:1 in Methods) would be helpful to the reader.

Response: We have clarified the text to be consistent with the 3:1 ratio. In addition, to the methods, we added the underlying basis for the ratio approach. This ratio was used to ensure similar coverage for lymphoid and myeloid cells across the 3 ccRCC patients, as myeloid cells tend to express a greater number of features.

3. The examination of the exhausted CD8+ T cell population found in the tumor was informative, revealing sub-populations with distinct profiles of exhaustion markers (Fig 2D). However, the authors' claim that these subsets have 'distinct functional properties' might be overstated, since no functional experiments were performed to validate this statement.

Response: Thank you for the suggestion and we agree, we have modified the language for better clarity and with limitation to the extent of what the data shows.

4. Please provide a color legend for 4D.

Response: Manuscript has substantially been changed with reviewer's suggestion; this is no longer relevant.

Reviewer #3 (Remarks to the Author):

Vishwakarma et al analyzed 24,904 lymphoid and myeloid cells in matched tumor and blood from 3 patients with clear cell renal cell carcinoma (ccRCC)by single-cell RNA sequencing using 10x genomics platform. both single-cell transcriptome and lymphocyte repertoire were analyzed. This is a descriptive report aimed to characterize ccRCC immune landscape. This is considered as the first report of the immune landscape of ccRCC using scRNA-seq. However, none of the major conclusions were validated by a secondary method.

Response: Towards validation, we have 1) performed highthroughput immunohistochemistry on the ccRCC samples showing enrichment of CD8+ and PD-1+ cells in tumor compared to adjacent normal tissue, 2) developed a signature to examine survival using bulk sequencing data in the Cancer Genome Atlas, 3) identifying proliferative CD8+ T cells as a poor prognostic marker, we used mass CyTOF data from Sade-Feldman et alia 2018 to identify cell the cell population in ccRCC in advanced-grade diseases.

Another major flaw of this study is the authors ignored the multiplet issue by drop-let based approach. "Cell suspensions were loaded onto a Chromium Single-Cell Chip along with the reverse transcription (RT) master mix and single cell 5' gel beads, aiming for 7,500 cells per channel." at this recovery rate, approximately 6% of the cells are multiplerts and the authors did not attempt to address how this would impact their analyses.

Response: We have clarified that cells were removed with UMIs > 5,000 to control for multiplerts, please see SCRS integration in the methods section.

Reviewers' comments:

Reviewer #1 (Remarks to the Author):

In the revised manuscript, authors added more scRNA-seq data from one healthy normal PBMC and three normal kidney tissue samples to have scRNA-seq data of 37,055 immune cells in total. Authors also validated the outcome prediction model based on machine-learning algorithm using TCGA ccRCC RNA-seq dataset. Authors conclude we can predict the clinical outcome of ccRCC with the immune signature from CD8+ T cells and TAM. Authors did not fully answer to the question on the immune suppression mechanism directed by those CD8+ T cells and TAM, which are too broad to specify the mechanism. To clarify the activity of CD8+ T cells or TAM in ccRCC, authors were asked to analyze the subtypes of CD8+ T cells and macrophages, which might give us a clue on the immune suppression mechanism such as increased population of exhausted CD8+ T cells. Because the number of cells analyzed in this manuscript is not many, authors can use public datasets to identify those subtypes of CD8+ T cells. In addition, we need to check whether those signatures related to ccRCC of CD8+ T cells and TAM is specific to ccRCC. To this end, authors were asked to test those signatures in other cancer types. Still revised manuscript does not provide scientific advancement in the field of tumor immunology.

Reviewer #2 (Remarks to the Author):

Zakharia and colleagues have significantly revised their original manuscript, incorporating additional single cell sequencing data of healthy peripheral blood and normal kidney tissue into their detailed characterization of the CD4+ T cell, CD8+ T cell, and myeloid compartments within clear cell renal cell carcinoma (ccRCC). The authors also identified CD8+ T cell and macrophage gene signatures that correlate with patient survival by training machine learning models on ccRCC data from TCGA. Overall, this revision presents a strengthened story of immune dynamics in ccRCC that sufficiently addresses the comments brought up by us and other reviewers.

Comments:

1. In addition to the presence of expanded clonotypes in NK cell clusters (as discussed by the authors), there also appears to be a nontrivial amount of T cell clones found in the monocyte and macrophage clusters (Figure 2A). Some discussion about potential reasons for this observation may be helpful for the reader.
2. The authors show that the CD8+ subcluster 6 and TAM subcluster 3 gene signatures are able to discriminate overall survival (Figure 6B and C). Does a model using a feature set combining these two gene signatures further improve performance? Perhaps this would suggest that the immune response in ccRCC is not driven solely by one cell type, but rather is coordinated across both lymphoid and myeloid responses.

Ansuman Satpathy
Joy Pai

Reviewer #3 (Remarks to the Author):

In this revised manuscript, the authors provided plenty new data to validate their scRNAseq findings using several other platforms. However, they did not fully address the multiplet issue. Multiplet identification tools such as Scrublet are recommended.

Reviewers' comments:

Reviewer #1 (Remarks to the Author):

In the revised manuscript, authors added more scRNA-seq data from one healthy normal PBMC and three normal kidney tissue samples to have scRNA-seq data of 37,055 immune cells in total. Authors also validated the outcome prediction model based on machine-learning algorithm using TCGA ccRCC RNA-seq dataset. Authors conclude we can predict the clinical outcome of ccRCC with the immune signature from CD8+ T cells and TAM. Authors did not fully answer to the question on the immune suppression mechanism directed by those CD8+ T cells and TAM, which are too broad to specify the mechanism. To clarify the activity of CD8+ T cells or TAM in ccRCC, authors were asked to analyze the subtypes of CD8+ T cells and macrophages, which might give us a clue on the immune suppression mechanism such as increased population of exhausted CD8+ T cells. Because the number of cells analyzed in this manuscript is not many, authors can use public datasets to identify those subtypes of CD8+ T cells. In addition, we need to check whether those signatures related to ccRCC of CD8+ T cells and TAM is specific to ccRCC. To this end, authors were asked to test those signatures in other cancer types. Still revised manuscript does not provide scientific advancement in the field of tumor immunology.

Extracted Points:

1. Authors did not fully answer to the question on the immune suppression mechanism directed by those CD8+ T cells and TAM, which are too broad to specify the mechanism. To clarify the activity of CD8+ T cells or TAM in ccRCC, authors were asked to analyze the subtypes of CD8+ T cells and macrophages, which might give us a clue on the immune suppression mechanism such as increased population of exhausted CD8+ T cells.

Response: As part of the resubmission and after incorporating additional single-cell data from normal kidneys and healthy peripheral blood controls, we performed the requested analyses on both CD8+ T cells (Figure 3) and TAM populations (Figure 5). For the CD8+ T cells, expression trajectory analysis found several distinct branches associated with exhaustion and proliferation (Figure 3E, G - enriched in the tumor CD8+ T cells) and independently associated with clonal expansion (Figure 3E, F). Specifically, CD8_0 was defined as “exhausted” T cells, with elevated expression of *CTLA4*, *TIM-3*, *PD-1* and *TIGIT* (Figure 3D) and were increased in tumor-infiltrating CD8+ T cells (Figure 3B). Further, we identified 5 distinct macrophage clusters, defining 3 of the 5 clusters as TAMs (Figure 5E) with several common and distinct gene expression patterns (Figure 5F) and gene set enrichment (Figure 5H). TAM_3 is particularly interesting due to its associated gene signature in prediction of worse prognosis (Figure 6B). TAM_3 subcluster exhibits an elevated *AREG* (Figure 5F), a known factor to enhance tumor-infiltrating regulatory T cells (PMID: 23333074; PMID: 27432879). Collectively these TAMs may maintain an immune suppressive/inactive microenvironment for human ccRCC.

2. In addition, we need to check whether those signatures related to ccRCC of CD8+ T cells and TAM is specific to ccRCC. To this end, authors were asked to test those signatures in other cancer types

Response: Thank you for the suggestions, we have applied gene signatures from ccRCC-trained k-nearest neighbor model on 33 cancer types in PANCAN TCGA (Supplemental Figure 8). We found that the CD8_6 signature is more broadly applicable to predict overall survival within 13 of 33 cancers and that the TAM_3 signature discriminated overall survival in only 3 cancer types. We don't fully understand the differential discrimination between CD8_6 and

TAM_3 signatures. Please note we did not train the gene signatures to all cancer types; nor did we intend to generalize these signatures for OS prediction among different cancer types.

3. Still revised manuscript does not provide scientific advancement in the field of tumor immunology.

Response: Here we want to highlight the technical and scientific advancements in relevant fields:

- 1) First and foremost, the manuscript is the first to combine single-cell gene expression and T cell receptor sequencing of immune cells in renal clear cell carcinoma.
- 2) In addition, the manuscript is the first tumor-based project to utilize the scRepertoire package we developed, allowing for assignment of clonotype by both TCRA and TCRB chains. This facilitates a more in-depth analysis for clonotype dynamics in both CD4+ and CD8+ T cells than other works in the single-cell tumor immunology field. We did find differential properties of CD4+ and CD8+ T cell clonality within blood and tumors. CD8+ T cells have significant overlapping clones between blood and tumors; whereas CD4+ T cells don't have (Figure 2).
- 3) Our work moves beyond just the characterization of a single-cell data set with the inclusion of multimodal points of data quantification – including bulk RNA sequencing and mass cytometry data sets, both of which point towards more generalizable trends in ccRCC oncoimmunology. This is all in addition to providing high-quality single-cell sequencing of paired peripheral blood and tumor-infiltrating immune cells from renal carcinoma patients to the field for further inquiry by tumor immunologists and other researchers. We have added some portions of this response to the discussion to highlight the novelty of the study.
- 4) We performed deep machine learning to extract gene signatures from all relevant tumor-infiltrating immune subtypes with CD8+ T cells and TAMs. We developed a 15-gene signature that can faithfully predict patient overall survival in ccRCC (Figure 6), with certain generalization to other immune-relevant cancer types (Supplemental Figure 8A-B).

Reviewer #2 (Remarks to the Author):

Zakharia and colleagues have significantly revised their original manuscript, incorporating additional single cell sequencing data of healthy peripheral blood and normal kidney tissue into their detailed characterization of the CD4+ T cell, CD8+ T cell, and myeloid compartments within clear cell renal cell carcinoma (ccRCC). The authors also identified CD8+ T cell and macrophage gene signatures that correlate with patient survival by training machine learning models on ccRCC data from TCGA. Overall, this revision presents a strengthened story of immune dynamics in ccRCC that sufficiently addresses the comments brought up by us and other reviewers.

Comments:

1. In addition to the presence of expanded clonotypes in NK cell clusters (as discussed by the authors), there also appears to be a nontrivial amount of T cell clones found in the monocyte and macrophage clusters (Figure 2A). Some discussion about potential reasons for this observation may be helpful for the reader.

Response: Thanks for raising the excellent point, we have added this discussion to the manuscript:

Page 13 “Clonotype assignment found single copy clonotypes and clonotypes with 1-5 copies distributed across myeloid clusters (Figure 2A). These cells are most likely T cells that cluster with myeloid cells as a result of partial loss of discriminating cell expression during integration across multiple sequencing runs, a major issue in single-cell analysis.”

In addition, we also performed doublet analysis and we found cluster 21 – which is close to B cell cluster 2 – had roughly 18% of doublets with recovered TCRs. We have also noted this discrepancy in the revision and included the doublet analysis (Supplemental Figure 2). Anecdotally, during the writing and usage of the scRepertoire package, recovered TCR in myeloid and B cells has been an issue we have encountered under other circumstances including autoimmune processes of human and mouse skin, peripheral blood in lymphoma, and lung tumors. Cluster 21 is excluded from further analysis in the manuscript.

2. The authors show that the CD8+ subcluster 6 and TAM subcluster 3 gene signatures are able to discriminate overall survival (Figure 6B and C). Does a model using a feature set combining these two gene signatures further improve performance? Perhaps this would suggest that the immune response in ccRCC is not driven solely by one cell type, but rather is coordinated across both lymphoid and myeloid responses.

Response: We appreciate the comment and we did believe immune cell interactions within the tumor microenvironment (see reviewer 1, question 1 as an example). With our experience in developing signatures, we have observed significant survival differentiation with 25 to 30 randomly assigned genes using the k-nearest neighbor model as well as other training models; whereas using less than 20 randomly selected genes unable to differentiate survival. Thus, combining the two 15-gene signatures would provide disingenuous conclusions. Instead, we have determined patient overlapping within different prognostic groups based on the two gene signatures. CD8_6 and TAM_3 signatures place 53 and 67 patients among the bad prognosis group, respectively, sharing overlapping 35 patients. The two signatures have 188 overlapping patients within the good prognosis groups, among 214 patients for CD8_6 and 200 for TAM_3. The significant overlapping in patients from both signatures indicate that combining these two signatures won't yield better performance than either single signature.

Pg20 “however, the CD8_6 and TAM_3 classifications general stratified the same patients either into good-prognosis (188 in both signatures) and poor-prognosis (35 in both signatures) groups (Fischer exact p value = 9.3e-15).”

pg23 “We note that the TAM_3 signature classification was significantly associated with the patients classified by the CD8_6 signature, suggesting the possible interaction or coordination between lymphoid and myeloid cells in ccRCC.”

Reviewer #3 (Remarks to the Author):

In this revised manuscript, the authors provided plenty new data to validate their scRNAseq findings using several other platforms. However, they did not fully address the multiplet issue. Multiplet identification tools such as Scrublet are recommended.

Response: We thank the reviewer for the suggestion, we have performed doublet/multiplet estimation using the scDbIFinder R package (Supplemental Figure 2). Based on the estimation, only a single cluster, *i.e.* cluster 21, had doublet estimation > 10%. This was a unique cluster, as it has recovered TCR, but clusters with B cells (cluster 2), possibly indicating a strong cell-cell interaction of B and T cells. Cluster 21 was excluded from all the downstream analysis of T cells. Other clusters in the integrated object had minimal doublet estimates, which is likely due

to the aforementioned filtering by number of unique features. With the further downstream analysis of CD8+, CD4+, and myeloid cells, no predominant doublet clusters were visualized. Percent of doublets in each subcluster were < 2%, with the exception of myeloid subcluster 12 with 4.87% doublets. Overall there is minimal effect of doublets/multiplet on the downstream analysis. We have added these points to the results and expanded the methods to reflect this new analysis.

REVIEWERS' COMMENTS:

Reviewer #3 (Remarks to the Author):

The Authors have addressed all my concerns.